# PV1 Protein from *Plasmodium falciparum* Exhibits Chaperone-Like Functions and Cooperates with Hsp100s

**DOI:** 10.3390/ijms21228616

**Published:** 2020-11-16

**Authors:** Kazuaki Hakamada, Manami Nakamura, Rio Midorikawa, Kyosuke Shinohara, Keiichi Noguchi, Hikaru Nagaoka, Eizo Takashima, Ken Morishima, Rintaro Inoue, Masaaki Sugiyama, Akihiro Kawamoto, Masafumi Yohda

**Affiliations:** 1Department of Biotechnology and Life Science, Tokyo University of Agriculture and Technology, Tokyo 184-8588, Japan; kazuaki.hakamada@yohda.net (K.H.); manami.nakamura@yohda.net (M.N.); rio.midorikawa@yohda.net (R.M.); k_shino@cc.tuat.ac.jp (K.S.); 2Instrumentation Analysis Center, Tokyo University of Agriculture and Technology, Tokyo 184-8588, Japan; knoguchi@cc.tuat.ac.jp; 3Division of Malaria Research, Proteo-Science Center, Ehime University, Ehime 790-8577, Japan; nagaoka.hikaru.ys@ehime-u.ac.jp (H.N.); takashima.eizo.mz@ehime-u.ac.jp (E.T.); 4Institute for Integrated Radiation and Nuclear Science, Kyoto University, Osaka 590-0494, Japan; morishima@rri.kyoto-u.ac.jp (K.M.); inoue.rintaro.5w@kyoto-u.ac.jp (R.I.); sugiyama.masaaki.5n@kyoto-u.ac.jp (M.S.); 5Institute for Protein Research, Osaka University, Osaka 565-0871, Japan; kawamoto@protein.osaka-u.ac.jp

**Keywords:** malaria, *Plasmodium falciparum*, translocon, chaperone unfolding, transport

## Abstract

*Plasmodium falciparum* parasitophorous vacuolar protein 1 (PfPV1), a protein unique to malaria parasites, is localized in the parasitophorous vacuolar (PV) and is essential for parasite growth. Previous studies suggested that PfPV1 cooperates with the *Plasmodium* translocon of exported proteins (PTEX) complex to export various proteins from the PV. However, the structure and function of PfPV1 have not been determined in detail. In this study, we undertook the expression, purification, and characterization of PfPV1. The tetramer appears to be the structural unit of PfPV1. The activity of PfPV1 appears to be similar to that of molecular chaperones, and it may interact with various proteins. PfPV1 could substitute CtHsp40 in the CtHsp104, CtHsp70, and CtHsp40 protein disaggregation systems. Based on these results, we propose a model in which PfPV1 captures various PV proteins and delivers them to PTEX through a specific interaction with HSP101.

## 1. Introduction

Malaria is caused by the protozoan parasite *Plasmodium*. *Plasmodium falciparum* is the most virulent species of this genus, infecting humans and causing serious problems worldwide [1]. These parasites invade erythrocytes, where they develop and reside in the parasitophorous vacuole (PV) [2]. Developing in the PV may enable the parasites to evade the human immune system. In the PV, malarial parasites produce and export numerous proteins to remodel the host erythrocytes [3,4,5,6]. A complex of proteins called *Plasmodium* translocon of exported proteins (PTEX) mediates the transport of proteins through the PV membrane (PVM) [7,8,9,10]. PTEX consists of EXP2, HSP101, PTEX150, PTEX88, and TRX2 [8,11]. Among these proteins, EXP2, HSP101, and PTEX150 are the essential components of PTEX for parasite survival [12], while PTEX88 and TRX2 are believed to play accessory roles [8]. EXP2 and PTEX150 are unique to malaria parasites [8]. Structural and electrophysiological studies have shown that EXP2 forms the pore that spans the PVM [11,13]. HSP101 belongs to the Hsp100/casein lytic proteinase (Clp) B family of proteins, which exist as homohexamers and mediate the refolding of proteins from aggregation with Hsp70/40 in an ATP-dependent manner [14,15,16]. Recent cryoEM structural analysis of the PTEX core complex has shown that EXP2 and PTEX150 interdigitate to create a static, funnel-shaped, pseudo-seven-fold-symmetric protein-conducting channel that spans the vacuolar membrane [11]. The spiral-shaped HSP101 hexamer is tethered above this funnel. Since the central pore of EXP2 is too narrow for the folded proteins, it is hypothesized that HSP101 unfolds these proteins and threads them through the PTEX pore. Supporting this hypothesis, inhibiting HSP101 induces the accumulation of exported proteins in the PV space [9].

Although PTEX serves to unfold exported proteins and thread them through into the host erythrocyte cytosol, it remains unknown how the exported proteins are selected. The *Plasmodium* export element (PEXEL) motif (RxLxE/Q/D) is exhibited by approximately 500 parasitic proteins and is considered to be the signal for PTEX-mediated export [17,18,19,20]. However, some PEXEL-negative proteins are also exported through PTEX [7].

*P. falciparum* parasitophorous vacuolar protein 1 (PfPV1; PF3D7_1129100), a protein that is unique to malaria parasites, is localized in the PV and is essential for parasite growth [21,22]. PfPV1 is also regarded as a PTEX accessory protein, since it coprecipitates with the PTEX complex and partially colocalizes with EXP2 [23,24,25]. In immunoprecipitation studies, PfPV1 was coimmunoprecipitated with exported proteins possessing a PEXEL motif, PfEMP1-trafficking protein (PTP5), and PF3D7_0801000 (unnamed protein) [23]. In contrast, the protozoan protein RAP1, which is not exported and localized in PV, was not observed to coimmunoprecipitate with PfPV1 [23]. In addition, the accumulation of PTP5 in PV was observed when the PfPV1 interaction site of PTP5 was deleted [23]. Exported proteins are not recognized only by the PEXEL motif in the PV space. Knocking down PfPV1 results in decreased levels of PfEMP1, KAHRP, EMP3, and other exported proteins that serve to modify the host cell by the protozoan after entry into the host cell [26]. Due to the reduced levels of these proteins, the modification of the host cell by the protozoa was clearly weakened [26].

These studies strongly suggest that PfPV1 is closely related to the protein export machinery PTEX and exported proteins. However, few studies have investigated the structural and functional characterization of PfPV1 in detail. In this study, we determined how PfPV1 is related to protein export by investigating the biochemical characterization of PfPV1.

## 2. Results

### 2.1. Expression and Purification of PfPV1

First, we attempted to express PfPV1 with a polyhistidine tag at the C-terminus, designated PfPV1-His, in *Escherichia coli*. Although PfPV1-His was overexpressed in the soluble fraction, this protein was difficult to purify by Ni^2+^-affinity chromatography (Appendix A). Surprisingly, various *E. coli* proteins were coeluted with PfPV1 in the imidazole fraction. This elution was almost the same as the supernatant of the lysate, and the same result was obtained repeatedly. Therefore, we suspect that PfPV1 has the ability to bind multiple proteins promiscuously. Next, PfPV1 was expressed as the fusion protein with glutathione S-transferase (GST-PfPV1). GST-PfPV1 was captured by glutathione sepharose affinity chromatography, and PfPV1 was subsequently eluted with digestion by HRV3C protease. The purity of the product was higher than that of PfPV1-His (Appendix A). The GST tag appears to inhibit the interaction of PfPV1 with other proteins due to steric hindrance or because of the oligomeric conformation changes. However, even further purification with various columns did not improve the purity further.

Finally, we expressed GST-PfPV1 with a Strep-tag at the C-terminus (GST-PfPV1-Strep). This protein, designated GST-PfPV1-Strep, was captured by a Strep-Tactin column. After the removal of the GST tag by HRV3C protease digestion, PfPV1-Strep was eluted from the column. SDS-PAGE analysis indicated that almost all other proteins were removed (Figure 1A). Next, the purified PfPV1-Strep was analyzed by native PAGE. The protein appeared as smear bands ranging from 146 to 1236 kDa (Figure 1B). Most PfPV1-Strep appeared to form large oligomers from 720 to 1236 kDa. The molecular mass of the smallest oligomer was approximately 146 kD, which corresponds to the trimer (monomeric PfPV1-Strep: 50.8 kDa). These observations suggested that PfPV1-Strep particles were not uniform immediately after elution.

Next, we attempted to obtain uniform PfPV1-Strep by in vitro refolding. The purified PfPV1-Strep was denatured in 4 M guanidine hydrochloride and was refolded by dialysis. Thus, the resulting refolded PfPV1-Strep (designated rPfPV1-Strep) appeared as a relatively small and uniform band in the native PAGE (Figure 1C).

### 2.2. Oligomer Structures of PfPV1-Strep and rPfPV1-Strep

The oligomeric structure of rPfPV1-Strep was analyzed by size exclusion chromatography–multiangle light scattering (SEC-MALS). Although there was still a void peak, which may have been the aggregates produced during the renaturation, rPfPV1-Strep primarily appeared as small oligomers with molecular masses of 217 and 140 kDa, probably corresponding to tetramers and trimers, respectively (Figure 2).

In electron microscopy (EM) images, PfPV1-Strep was observed to exist as large heterogeneous particles of 20–30 nm, which coincided with the native PAGE analysis (Figure 3A). A 2D classification showed elongated structures. In contrast, rPfPV1-Strep was observed to exist as relatively uniform particles smaller than 10 nm (Figure 3B). A 2D classification of the images exhibited rhombus tetrameric structures (Figure 3C). The trimers indicated by native PAGE and SEC-MALS were not observed, probably due to the stability or the arrangement of the structure.

Next, PfPV1-Strep and rPfPV1-Strep were analyzed by sedimentation velocity analytical ultracentrifugation (SV-AUC). rPfPV1-Strep was observed to primarily exist as a trimer and tetramer. In addition, peaks for the monomer and larger oligomers were observed (Figure 4A, Table 1). PfPV1-Strep was observed as various oligomeric structures (Figure 4B, Table 1). The peaks at 221 and 139 kDa may correspond to tetramers and trimers, respectively, for rPfPV1-Strep. In addition, monomers, dimers, and larger peaks were observed in PfPV1-Strep. Interestingly, the average molecular mass difference between the peaks from 222 to 2267 kDa was 227.2 kDa, corresponding to the rPfPV1-Strep tetramer. This result suggests that PfPV1-Strep primarily forms large oligomers that are assembled from the stable tetramers observed in EM.

### 2.3. Chaperone-Like Activity of PfPV1 for Dithiothreitol (DTT)-Induced Insulin Aggregation

Our observations indicated that PfPV1 has the ability to interact promiscuously with various proteins. We believed that PfPV1 may interact with various proteins, such as molecular chaperones, enabling it to transfer plenty of proteins to PTEX. The HSP100 family of chaperones, to which PfHSP101 belongs, cooperates with other chaperones, Hsp70 and Hsp40 [14,15,16]. These proteins capture substrates and transfer them to Hsp100. However, there is no report describing the cooperation of PfHSP101 with chaperones. Therefore, we examined the general chaperone-like ability of rPfPV1-Strep by the dithiothreitol (DTT)-induced aggregation assay of insulin. Through treatment with DTT, insulin is denatured by the reduction of the disulfide bond and forms aggregates [27]. Insulin was treated by DTT with or without rPfPV1-Strep, and aggregation formation was observed by measuring absorbance at 360 nm. The addition of rPfPV1-Strep suppressed the aggregation of insulin in a concentration-dependent manner (Figure 5). The results show that rPfPV1-Strep can protect the reduced insulin from aggregating. These findings suggest that rPfPV1 has the ability to bind to proteins, including unstable proteins or proteins with exposed hydrophobic surfaces, and keep them soluble, similar to chaperones.

### 2.4. Interaction of PfPV1 with PTEX Core Proteins

Then, we examined the interaction of PfPV1 with PTEX core components by surface plasmon resonance. PfPV1, PTEX core proteins, namely, PfEXP2, PfPTEX150, and PfHSP101, and a control protein, *P. falciparum* reticulocyte-binding protein homolog 5 (Rh5) [28], were expressed as GST fusion proteins by the wheat germ in vitro protein expression system and purified by a glutathione Sepharose 4B column. GST-PV1 was immobilized on a CM5-Chip, and GST-EXP2, GST-PTEX150, GST-HSP101, and GST-Rh5 were employed as analytes. Kinetics analysis of GST-PfPV1 between GST-EXP2, GST-PTEX150, and GST-HSP101 exhibited interactions with K_D_ values of 1.10 × 10^−7^, 5.46 × 10^−8^, and 1.05 × 10^−8^ at 1:1 stoichiometry, respectively (Figure 6 and Table 2). Among these interactions, the interaction between GST-HSP101 and GST-PfPV1 was the strongest. In the cryoEM structure, EXP2 was observed to form the transmembrane pore and HSP101 was determined to be connected to it by PTEX150 [11]. Substrate proteins should be unfolded by HSP101 and transferred through the PTEX150 and EXP2 inner pores. Thus, it is reasonable to speculate that PfPV1 may have a specific affinity to HSP101 to hand over various proteins. Rh5 is localized to the rhoptories and plays a central role in the merozoite invasion [29]; thus, Rh5 is not related with PTEX. As expected, almost no interaction between GST-Rh5 and GST-PV1 was observed.

### 2.5. Cooperation with the Hsp104 System

Our results indicated that PV1 directly interacts with HSP101 and suppresses protein aggregation, but it is unknown how PV1 participates in PTEX-mediated protein export. HSP101 is the homolog of Hsp104/ClpB, which captures aggregated or unstable proteins and unfolds them in an ATP-dependent manner. Generally, the disaggregation mechanism employed by Hsp104/ClpB requires Hsp70 and Hsp40 to serve as cofactors. Therefore, HSP101 should have such chaperone partner(s) if it is essential for the unfolding mechanism of exported proteins by the PTEX complex.

We attempted to express and purify PfHSP101 as a recombinant protein by *E. coli* or the wheat germ in vitro protein expression system to study the disaggregation activity of PfHSP101. However, the full-length PfHSP101 without solution tags, such as the GST-tag, appeared in the insoluble fraction. This result is probably due to the original nature of HSP101, which forms a complex with other PTEX core components, PTEX150 and EXP2.

Next, we attempted to examine the cooperation of rPfPV1-Strep with the Hsp104 system from *Chaetomium thermophilum*, CtHsp104 (manuscript in preparation). CtHsp104 can refold aggregated firefly luciferase with Hsp70 and Hsp40 from *C. thermophilum*, CtHsp70 and CtHsp40 (Appendix A). We examined the effect of the substitution of the component by rPfPV1-Strep on disaggregation activity. Interestingly, PfPV1 could almost completely substitute for CtHsp40. The mixture of CtHsp104, CtHsp70, and PfPV1 exhibited an approximately 80% equivalent activity compared with the original Hsp100 system (Figure 7). Hsp40s possess unique classes of polypeptide-binding domains that bind and deliver specific clients to Hsp70s. Hsp40 also contains a highly conserved J domain, by which Hsp40 interacts with Hsp70 and accelerates its ATP hydrolysis activity [30,31]. Although there is no J domain in PfPV1, it may cooperate with PfHSP101 to unfold the exported proteins (Appendix A).

## 3. Discussion

In this study, we performed functional and structural characterizations of PfPV1. Interestingly, multiple proteins nonspecifically bound to PfPV1 when PfPV1 was expressed in *E. coli*. When GST was added to the N-terminus, this nonspecific interaction was suppressed and it was possible to purify the protein by using Strep-tag. Purified PfPV1-Strep formed various oligomeric structures. By ultracentrifugation analysis, PfPV1-Strep was observed to form oligomers with various molecular weights in addition to monomers, dimers, trimers, and tetramers. Since the difference in molecular mass is approximately 227.2 kDa, it is plausible that the tetramers assembled to form large oligomers.

On the other hand, rPfPV1-Strep, obtained by unfolding, diluting, and refolding PfPV1-Strep, forms relatively small oligomers, trimers, and tetramers. Electron microscopy analysis showed that this protein has a rhombic tetrameric structure. Therefore, we believe that the rhombic tetrameric structure is the structural unit of PfPV1, which assembles into various oligomers. It remains unclear whether the oligomeric structure of PV1 is present in nature. However, since tetramers can be classified by EM images (trimers could not be classified), that state is able to exist stably and consists of the smallest unit of a large oligomer.

Because rPfPV1-Strep could prevent insulin from aggregating and GST-tagged PV1 may mask its hydrophobic region, rPfPV1-Strep interacts with the client protein with hydrophobic interactions. Additionally, since there are no common motifs among the exported proteins that have been observed to be involved with PfPV1, PfPV1 may promiscuously bind to the exported proteins hydrophobically. Hsp104/ClpB has a hexameric ring structure and unfolds protein aggregates in cooperation with Hsp70/Hsp40. Hsp104/ClpB is a member of the ATPases associated with diverse cellular activities (AAA+ proteins) superfamily [32,33] and consists of an N-terminal domain (ND), an M-domain (MD), and two AAA+ modules (AAA1 and AAA2) [34]. Each AAA+ module binds and hydrolyses ATP and causes structural changes required for protein disaggregation [35,36,37,38,39]. PfHSP101 nucleotide binding domains (NBDs) share approximately 40% and 39% amino acid sequence identity with *E. coli* ClpB and yeast Hsp104, respectively [40]. HSP101 is exported PV space by the ER signal sequence and assembles with EXP2 and PTEX150 to form PTEX. HSP101 is believed to unfold proteins to enable them to pass through the central pore of PTEX. To characterize PV1 involvement in HSP101, we attempted to express HSP101 in *E. coli*. However, the protein was obtained as aggregates. Next, we attempted to examine the cooperation of PfPV1 with the Hsp104-70-40 system in disaggregation and found that PfPV1 substitutes the function of Hsp40. Hsp40/DnaJ is a co-chaperone of Hsp70/DnaK containing a highly conserved J domain that binds denatured proteins and stimulates the ATPase activity of Hsp70/DnaK (Appendix A) [41,42].

Several Hsp70 homologs exist in *P. falciparum* [43]. Among these proteins, only Hsp70-x localizes both in the PV and infected erythrocyte cytosol. Hsp70-x may be involved in maintaining proteins in an unfolded state prior to passage across PTEX in combination with HSP101 and folding of PV resident proteins. Chemical crosslinking followed by immunoprecipitation using anti-Hsp70-x specific antisera and immunoblotting using specific antisera has shown that Hsp70-x is incorporated into a complex or complexes containing PFE55GFP, PfGBP130, PfPHIST_0801, PfHSP101, PfHsp70, PfPV1, and PfExp2. Hsp40 was not included in the complexes [44]. However, we must abandon the idea that PfPV1 cooperates with PfHsp70x because Hsp70-x is dispensable for the *Plasmodium falciparum* intraerythrocytic life cycle [45]. Sequence alignment of HSP101 and Hsp104 shows that the primary difference is located in the M domain region. The M domain of Hsp104 is regarded as the interaction site for Hsp70 to regulate Hsp104 function. Therefore, HSP101 might function without the support of Hsp70. In that case, PfPV1 would capture multiple proteins and deliver them to HSP101 of PTEX in PV. Next, HSP101 would unfold and thread them through the pores of PTEX150 and EXP2. The exported proteins would be refolded by molecular chaperones in the host cell cytosol. A previous study showed that TCP-1 Ring Complex (TRiC)/Chaperonin containing TCP-1 (CCT) might be involved in the folding of exported proteins [26]. Our model is presented in Figure 8. PfPV1 captures various proteins by interacting with the exposed hydrophobic region and hands them over to HSP101 of PTEX. HSP101 unfolds them and threads them through the pore of PTEX. The transported proteins are refolded by the action of molecular chaperones, including TRiC/CCT.

## 4. Materials and Methods

### 4.1. Expression and Purification of PfPV1

To express PfPV1-His, the *pfpv1* fragment encoding Val-23 to Ser-453 was amplified from *pfpv1* cDNA from *Plasmodium falciparum* 3D7 and cloned into the NdeI/XhoI of pET28b. GST-PV1 and GST-PV1-Strep were expressed using the pGEX-6p-1 vector. The *pfpv1* fragment encoding Val-23 to Ser-452 was amplified from *pfpv1* cDNA and cloned into the BamHI/NotI site. The primers used for amplification are shown in Appendix A. GST-PV1-Strep was expressed and purified as follows. *E. coli* BL21(DE3), transformed with the constructed plasmid, was grown in LB media with ampicillin at 37 °C. At OD600 0.5, 0.1 mM IPTG was added and incubated further at 25 °C for 24 h. *E. coli* cells were disrupted by sonication in 25 mL of binding buffer (50 mM Tris-HCl (pH 8.0), 200 mM NaCl, 1 mM DTT, and 1 mM EDTA). After the removal of precipitants by centrifugation at 14,000 g for 25 min, the supernatant was collected. After the addition of 1% CHAPS, the supernatant was filtered through a 0.45-μm membrane (25CS, ADVANTEC Co., Tokyo, Japan). The filtered supernatant was applied on a Strep-Tactin^®^ Sepharose column (5 mL) equilibrated with binding buffer with 1% CHAPS. Next, the column was washed with 50 mL of binding buffer with 1% CHAPS and 50 mL of binding buffer. To remove the GST tag, 7 mL of binding buffer containing 70 μL of Human rhinovirus (HRV) 3C Protease (1 U/μL, Takara) was applied to the column and kept at 4 °C for overnight. After washing with 50 mL of binding buffer, PfPV1Strep was eluted with 10 mL of elution buffer (50 mM Tris-HCl (pH 8.0), 200 mM NaCl, 2.5 mM D-desthiobiotin, 1 mM DTT, and 1 mM EDTA).

### 4.2. Refolding of PfPV1-Strep

PfPV1 was mixed with an equal volume of 8 M guanidine hydrochloride and incubated for 30 min at 4 °C. Then, the unfolded PfPV1-Strep was refolded by dialysis against more than 1000 times the volume of dialysis buffer (20 mM sodium phosphate (pH 6.8) and 200 mM NaCl) at least twice. Then, PfPV1-Strep and the refolded one, rPfPV1-Strep, were analyzed by SDS-PAGE and 4–16% Tris-glycine native PAGE.

### 4.3. Size Exclusion Chromatography–Multiangle Light Scattering (SEC-MALS)

The purified rPfPV1-Strep was analyzed by size exclusion chromatography by a KW-804SQ column (Showa Denko, Tokyo, Japan) connected with a UV detector (SPD-10AVp; Shimadzu, Kyoto, Japan), a multiangle light-scattering detector (MINI DAWN; Wyatt Technology, Santa Barbara, CA, USA), and a differential refractive index detector (Shodex RI-101; Showa Denko) using an HPLC system, PU-980i (JASCO, Tokyo, Japan), as described previously [46]. A 100-μL aliquot of sample was injected into the column and eluted with 20 mM sodium phosphate buffer (pH 6.8) and 200 mM NaCl at 0.5 mL/min. The molecular weight and protein concentration were determined according to the instructional manual (Wyatt Technology).

### 4.4. Electron Micrograph and Image Processing

Samples were applied to carbon-coated copper grids and negatively stained with 2.5% (w/v) uranyl acetate. Micrographs were recorded at a magnification of 50,000× with a JEM-1400 transmission electron microscope (JEOL, Tokyo, Japan) operated at 120 kV. Approximately 300 particles were manually picked from 2 micrographs and subjected to reference-free 2D classification for making templates using Relion-3.0 [47]. In total, 25,657 particle images were automatically picked up from 48 micrographs, and then 2D classification was performed using Relion-3.0.

### 4.5. Analytical Ultracentrifugation

Sedimentation velocity analytical ultracentrifugation (SV-AUC) measurements were carried out with ProteomeLab XL-I (Beckman Coulter, Brea, CA, USA). Samples were filled in 12-mm pathlength Epon double sector centerpieces. All measurements were performed using absorbance optics at a 40,000-rpm rotor speed at 25 °C. The time evolution of sedimentation data was analyzed with the Lamm formula, and then the weight concentration distribution of components, c(s20,w), was obtained as a function of the sedimentation coefficient. Here, the sedimentation coefficient was normalized to be the value at 20 °C in pure water, s20,w. The molecular weight M of each component was calculated using the corresponding peak value s20,w and the friction ratio f/f0. These calculations were performed with SEDFIT software [48]. In addition, the weight fraction w for each component was derived from the peak area estimated by the Gaussian function. The characteristic parameters are summarized in Table 1.

### 4.6. Insulin Aggregation Assay

The insulin aggregation assay was performed according to the method described previously with a slight modification [49,50]. Two milligrams of human insulin (FUJIFILM Wako Chemicals, Osaka, Japan) was dissolved in 1 mL of 10 mM HCl and kept at −20 °C. The assay mixture was prepared by mixing 150 µL of acid-denatured insulin (2 mg/mL) and 750 µL of 100 mM sodium phosphate buffer (pH 6.8) containing 400 mM NaCl and 30 µL of 20 mM DTT, rPfPV1-Strep, and H_2_O up to 1.5 mL in a quartz cell. The absorbance was measured by observing absorbance at 360 nm at 25 °C using a UV-VIS spectrophotometer (V-650, JASCO, Tokyo, Japan).

### 4.7. Surface Plasmon Resonance

PfPV1, EXP2, PTEX150, HSP101, and Rh5 were expressed as GST fusion proteins by the wheat germ in vitro protein expression system. The *pfpv1* fragment encoding Val23 to Ser453, the *ptex150* fragment encoding Glu-130 to Asn-993, and the *hsp101* fragment encoding Ala-27 to Pro-809 were amplified from cDNA derived from *P. falciparum* 3D7 and cloned into XhoI and NotI and then ligated into pEU-E01-GSTTEV-N2, specifically designed for the wheat germ cell-free protein expression system (CellFree Sciences, Matsuyama, Japan). The *exp2* fragment encoding Asp-25 to Glu-287 and the *rh5* fragment encoding Glu-26 to Gln-526 were also amplified and cloned into the XhoI/BamHI site of pEU-E01-GSTTEV-2. The proteins were expressed with Wheat germ cell-free protein expression system (CellFree Sciences) as N-terminal glutathione S-transferase (GST) fusion proteins [51]. The expressed proteins were purified by affinity chromatography using glutathione Sepharose 4B (Cytiva, Tokyo, Japan).

Surface plasmon resonance (SPR) experiments were performed using a Biacore X100 instrument (Cytiva) according to the manufacturer’s instructions. Biacore X100 evaluation software was used for single-cycle kinetic analysis. Sensor chip CM5, amine coupling reagents, and buffers were purchased from Cytiva. Fresh HBS-EP + (10 mM HEPES, pH 7.4, 150 mM NaCl, 3 mM EDTA, 0.05% (*v*/*v*) surfactant P20) was used as a running buffer for all SPR experiments. A blank flow cell was used to subtract buffer effects on the sensorgrams. After subtraction of the contribution of the bulk refractive index and nonspecific interactions with the CM5 chip surface, individual association (*k_a_*) and dissociation (*k_d_*) rate constants were obtained by global fitting of the data. Measurement conditions were optimized such that the contribution of mass transport to the observed values of *K_D_* was negligible.

### 4.8. Luciferase Disaggregation Assay Using the Hsp104 System

A disaggregation assay using firefly luciferase with Hsp104, Hsp70, and Hsp40 from *C. thermophilum* was performed with slight modification as described in our previous study (manuscript in preparation). Firefly luciferase (Sigma, St. Louis, MO, USA) was dissolved in unfolding buffer (25 mM HEPES, 50 mM KCl, 5 mM MgCl_2_, 5 mM 2-mercaptoethanol, and 4 M urea, or no urea for the positive control) and incubated at 30 °C for 30 min with small agitation. The final concentration of the treated luciferase was 3.125 µg/mL. To disaggregate luciferase, 41.33 µL of refolding buffer (25 mM HEPES, 50 mM KCl, 5 mM MgCl_2_, 1 mM DTT, and 1 mM ATP containing 1.2 µM each combination of rPfPV1-Strep or chaperone proteins) was added to 0.33 µL of aggregation sample. After 30 min of incubation at 30 °C, the detection reagent containing luciferin was added to the samples with 2 min of incubation, and luminescence was measured by a GloMax^®^ 20/20 luminometer (Promega, Madison, WI, USA). The degree of refolding was calculated by the standard sample added refolding buffer without proteins to undenatured luciferase.

## Figures and Tables

**Figure 1 ijms-21-08616-f001:**
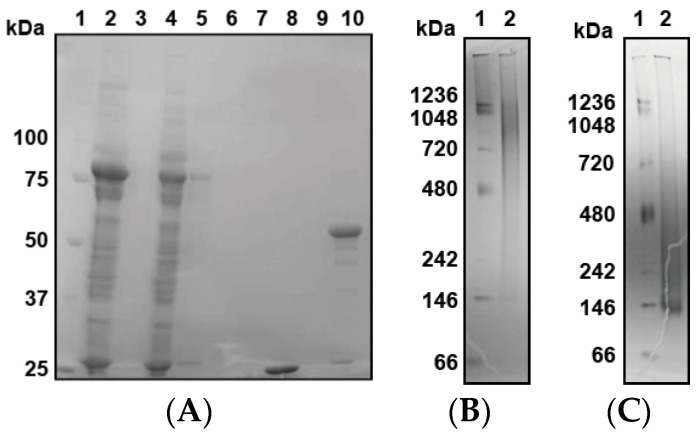
Expression, reconstitution, and purification of PfPV1-Strep. (**A**) SDS-PAGE analysis of GST-PfPV1-Strep expression in *Escherichia coli* and purification (1. Marker; 2. Supernatant of cell lysate; 3. Pellet of cell lysate; 4. Flow-through fraction of Strep-Tactin column; 5. Wash (with 1% 3-[(3-Cholamidopropyl)dimethylammonio]propanesulfonate (CHAPS)) fraction; 6. Wash fraction; 7. Bing buffer containing HRV3C protease; 8. Flow through after digesting glutathione S-transferase (GST)-tag; 9. Wash fraction; 10. Elution of PfPV1-Strep). (**B**) Native PAGE analysis of PfPV1-Strep (1. Marker; 2. PfPV1-Strep). (**C**) Native PAGE analysis of rPfPV1-Strep (1. Marker; 2 rPfPV1-Strep).

**Figure 2 ijms-21-08616-f002:**
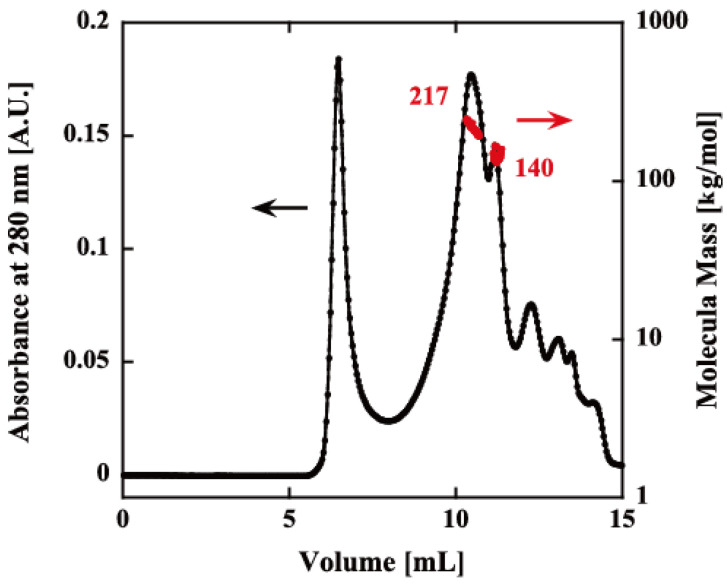
Size exclusion chromatography–multiangle light scattering (SEC-MALS) of rPfPV1-Strep. The refolded protein was analyzed using an SEC column (KW-804SQ). Chromatograms display the refractive index (black) with the molar mass of peaks determined by MALS (red). The first peak after ~5 mL is the void peak. The second (217 kDa) and third peaks (140 kDa) indicate the rPfPV1-Strep oligomers. The other peaks downstream were small molecules.

**Figure 3 ijms-21-08616-f003:**
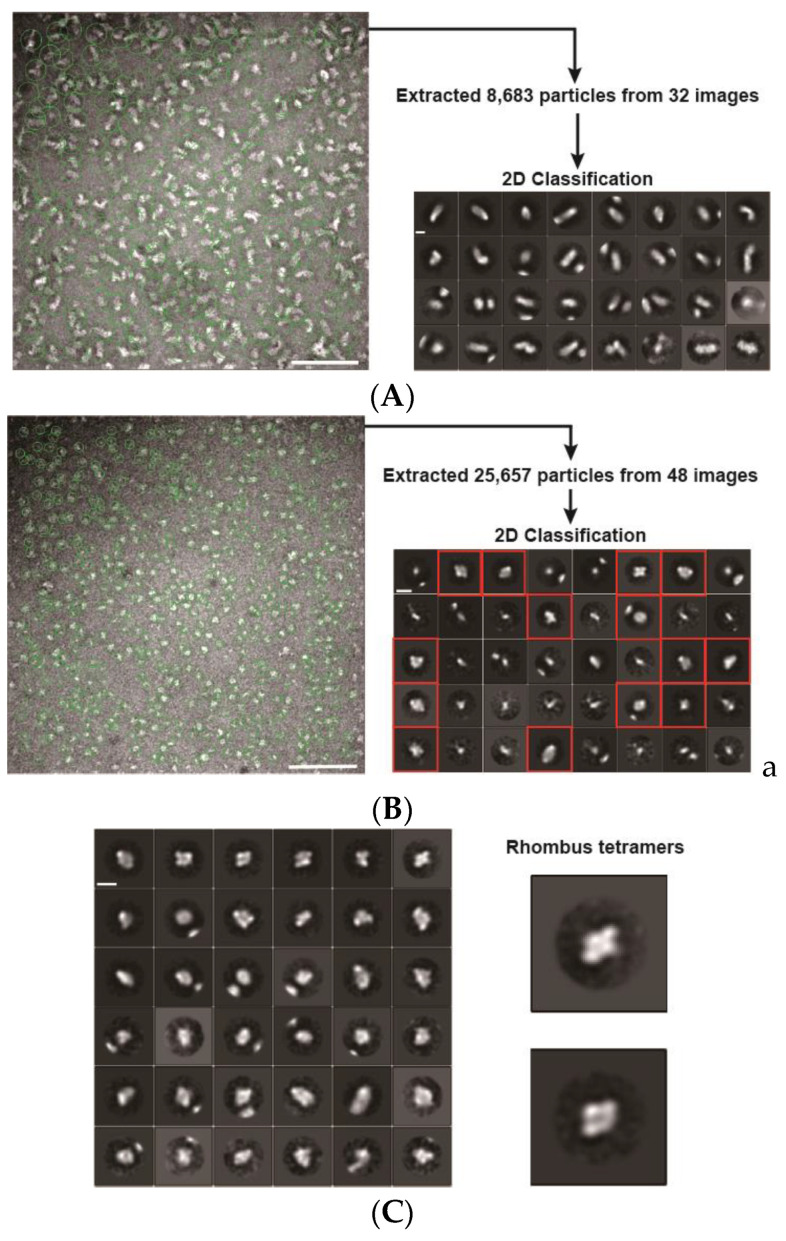
Electron microscopy (EM) images and 2D classifications. (**A**) EM image and 2D classification of PfPV1-Strep; (**B**) EM image and 2D classification of rPfPV1-Strep; (**C**) second 2D classification of rPfPV1-Strep. A total of 9228 particles of red boxes of 2D classification in (**B**) were used. Enlarged images of rhombus tetramers are shown. Scale bars represent 100 nm in the original images and 10 nm in the enlarged images.

**Figure 4 ijms-21-08616-f004:**
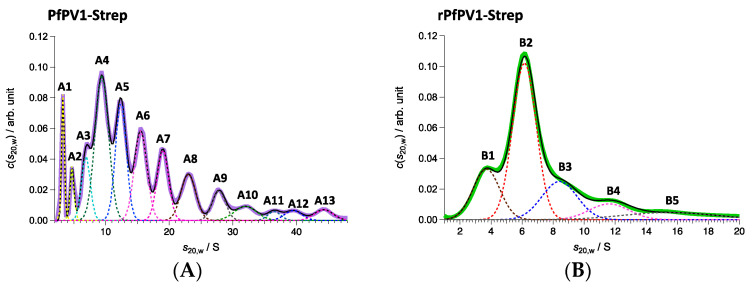
Sedimentation velocity analytical ultracentrifugation (SV-AUC) analyses of PfPV1-Strep (**A**) and rPfPV1-Strep (**B**). (**A**) Solid purple line represents *c*(*s*_20,w_) obtained with SV-AUC. Solid black line shows the result of the least square fitting with the summation of 13 Gaussian curves each of which corresponds to the components A1–A13: The dotted lines also express the fitting lines for the components and the peak potions are marked with the labels of A1–A13. PfPV1- Strep has high polydispersity. A protein series was detected in the range of 46–2267 kDa with 222 kDa as the main constituent factor. (**B**) Solid green line represents *c*(*s*_20,w_) obtained with SV-AUC. Solid black line shows the result of the least square fitting with the summation of 5 Gaussian curves each of which corresponds to the components B1–B5: The dotted lines also express the fitting lines for the components and the peak potions are marked with the labels of B1–B5. The main proportion, sharing 48.1 w/%, was 141 kDa corresponding to trimers. Almost the same amount of oligomers (59 kDa corresponding to monomers, 214 kDa corresponding to tetramer) were detected additionally.

**Figure 5 ijms-21-08616-f005:**
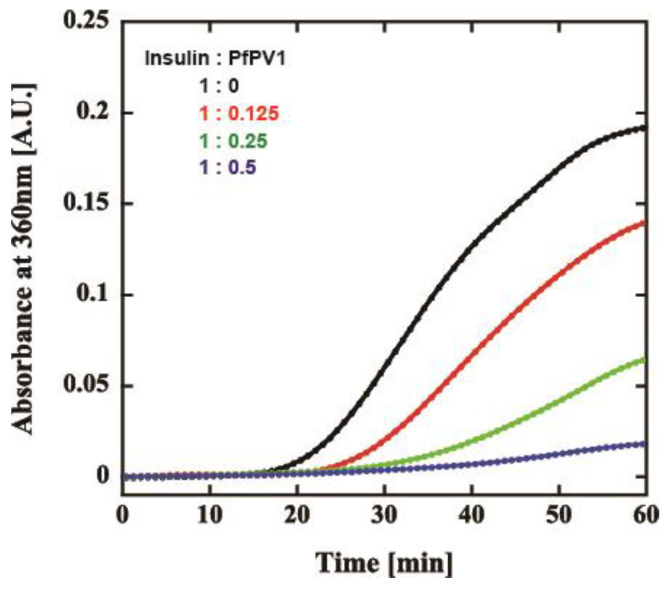
Effect of rPfPV1-Strep on the aggregation of insulin. Dithiothreitol (DTT)-induced insulin aggregation in the presence of rPfPV1-Strep at weight ratios of 1:0 (black), 1:0.1 (red), 1:0.25 (green), and 1:0.5 (blue) was monitored at an absorbance of 360 nm.

**Figure 6 ijms-21-08616-f006:**
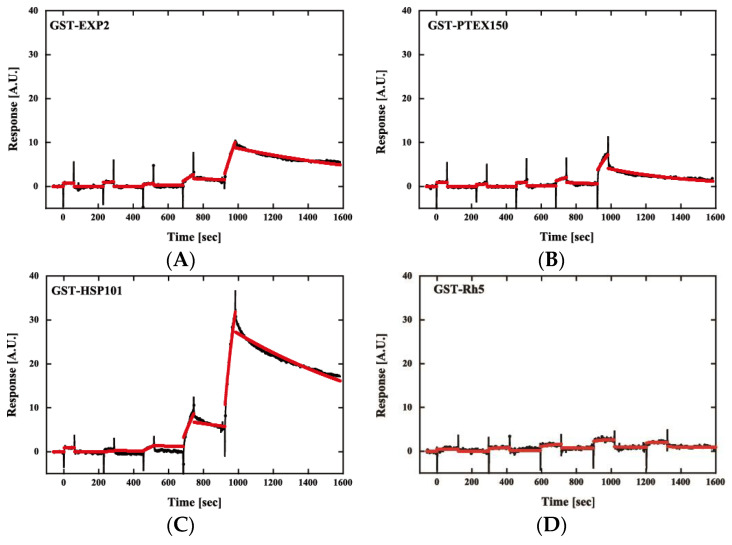
Sensorgrams of affinity measurements between GST-PfPV1 and GST-tagged *Plasmodium* translocon of exported proteins (PTEX) components. GST-PfPV1 was immobilized on the sensor chip, and GST-tagged PTEX components GST-EXP2 (**A**), GST-PTEX150 (**B**), and GST-HSP101 (**C**) were used as analytes. GST-Rh5 was used as the negative control (**D**). The analyte concentrations were 6.25, 12.5, 25, 50, and 100 nM. The sensorgrams (black) were fitted with single-cycle kinetic analysis (red).

**Figure 7 ijms-21-08616-f007:**
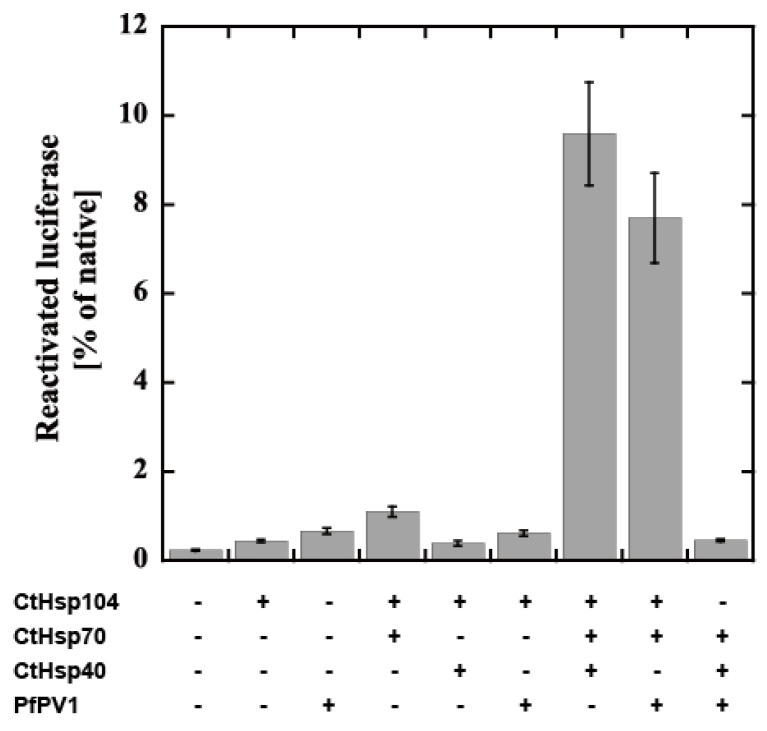
Reactivation of luciferase by the CtHsp104 system and PfPV1. Luciferase was denatured by urea treatment and applied for reactivation with the various combinations of CtHsp104 system components and rPfPV1-Strep. The bioluminescence of reactivated luciferase is expressed as the relative value of untreated luciferase.

**Figure 8 ijms-21-08616-f008:**
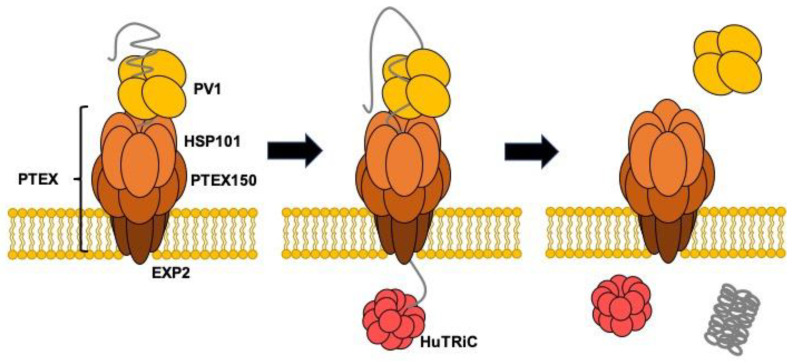
Schematic model for the role of PV1 in the protein transport by PTEX.

**Table 1 ijms-21-08616-t001:** Parameters of the components of SV-AUC.

PfPV1-Strep			
Component	*s*_20,w_/S	*M*/kDa	*w*/%
A1	3.3	46	4.2
A2	4.7	79	2.6
A3	6.9	139	6.8
A4	9.4	222	24
A5	12.4	336	15.7
A6	15.5	469	13.7
A7	19.1	637	9.9
A8	23.0	848	8.4
A9	27.8	1127	4.8
A10	31.9	1392	3.9
A11	36.5	1696	1.4
A12	39.5	1909	2.1
A13	44.3	2267	2.5
**rPfPV1-Strep**			
B1	3.6	59	17.6
B2	6.2	141	48.1
B3	8.4	215	17.5
B4	11.6	340	8.7
B5	15.5	539	8.1

**Table 2 ijms-21-08616-t002:** Kinetic parameters for the interaction of PfPV1 with PTEX core proteins.

	*k*_a_ (1/Ms)	*K*_d_ (1/s)	*K*_D_ (M)
GST-Rh5	ND	ND	ND
GST-EXP2	9.03 × 10^3^	9.89 × 10^−4^	1.10 × 10^−7^
GST-PTEX150	3.73 × 10^4^	2.04 × 10^−3^	5.45 × 10^−8^
GST-HSP101	8.70 × 10^4^	9.14 × 10^−4^	1.05 × 10^−8^

ND: not determined.

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
