# Peer review of "PV1 Protein from *Plasmodium falciparum* Exhibits Chaperone-Like Functions and Cooperates with Hsp100s"

_ijms, 2020, doi:10.3390/ijms21228616_

Round 1

Reviewer 1 Report

This Study "PV1 protein from Plasmodium falciparum exhibits chaperon-like functions and cooperates with Hsp100s" presented by Hakamada K et al. is well written and executed. I have some doubts:

  1. In Fig 2. X-axis unit is volume (ml) but in figure legend author said after "5 min". is it in time unit or volume (clarify).

Author Response

Thank you very much for your important comment. The answer to your query is as follows. The changes in the text are marked by red font in the revised manuscript.

Q: In Fig 2. X-axis unit is volume (ml) but in figure legend author said after "5 min". is it in time unit or volume (clarify).

A: We are sorry for our careless mistake. The description in the legend was wrong. It was corrected.

Reviewer 2 Report

The study aims to express, purify and biochemically characterize PfPV1, a parasitophorous vacuolar protein in Plasmodium falciparum.  The authors show that recombinantly expressed PV1 show activity which is similar to chaperones and interacts with members of the PTEX system. They further propose that PfPv1 captures proteins in the parasitophorous vacuole and deliver them to the PTEX system through interaction with HSP101. The manuscript provides interesting findings. However, there are some concerns.

  1. The authors suspect that PfPV1 has the ability to bind multiple protein promiscuously. In Figure 6 the authors show that GST-PfPV1 binds to members of the PTEX complex. I think it is necessary to show that the protein does not bind to other Plasmodium proteins promiscuously. I will prefer another Plasmodium GST-tagged protein which is not a member of the PTEX complex as control.
  2. There is a mix up in the Figure 4. Which of the graphs is PfPV1-STREP and rPfPV1-strep because the legend is contradictory. This also has to be corrected in the text.
  3. Figure 2 axis, Molecular Mass not Molecula mass
  4. Table 2 heading: “of” is omitted
  5. In the entire manuscript the authors use mL and ml and µL and µl interchangeably. This should be homogenous.

Author Response

Thank you very much for your important comments. The answers to your queries are as follow. The changes in the text are marked by red font in the revised manuscript.

Q: The authors suspect that PfPV1 has the ability to bind multiple protein promiscuously. In Figure 6 the authors show that GST-PfPV1 binds to members of the PTEX complex. I think it is necessary to show that the protein does not bind to other Plasmodium proteins promiscuously. I will prefer another Plasmodium GST-tagged protein which is not a member of the PTEX complex as control.

A: Thank you very much for your important suggestion. According to your suggestion, we have conducted surface plasmon experiments for the interaction of PfPV1 with Rh5, which is not exported by PTEX. As shown in the revised Fig. 6, no interaction was observed. The related description is included in the text.

Q: There is a mix up in the Figure 4. Which of the graphs is PfPV1-STREP and rPfPV1-strep because the legend is contradictory. This also has to be corrected in the text.

A: We are sorry for our careless mistake. The legend was mistaken. It was revised.

Q: Figure 2 axis, Molecular Mass not Molecula mass

A: We are sorry for our careless mistake. It was corrected.

Q: Table 2 heading: “of” is omitted

A: We are sorry for our careless mistake. It was corrected.

Q: In the entire manuscript the authors use mL and ml and µL and µl interchangeably. This should be homogenous.

A: Thank you very much for your kind comment. We checked thoroughly.

Reviewer 3 Report

The manuscript presented by Hakamada and collaborators presents the biochemical characterization of a unique malaria parasite protein PfPV1, which role in the protein transport by PTEX is proposed. The work revealed a true methodological challenge in order to get purified PfPV1 which they have solved by unfolding, diluting and refolding PfPV1, where the authors clearly presents its limitations and results interpretation caution. The work is very well presented and the data discussed properly.

A minor suggestion:

-Figure 3A and 3B requires a scale bar description and figure 3C requires a scale bar.

Author Response

Thank you very much for your important comments. The answers to your query is as follows. The changes in the text are marked by red font in the revised manuscript.

Q: Figure 3A and 3B requires a scale bar description and figure 3C requires a scale bar.

A: Thank you very much for your important comment. There were mistakes in scale bars. We revised them and gave descriptions.

Round 2

Reviewer 2 Report

The authors have used the Rh5-GST as control but in Figure 6 and Table 2 the information has not been revised. Line 185-186, they still write GST-His was used as negative control for figure 6D and there is no kinetic parameter on Table 2.

Author Response

We are sorry for our careless mistakes. We revised the legend of Figure 6 and Table 2. The changes are shown in red font. We sincerely appreciate for your review.